

# Exploring the climate system response to a range of freshwater representations: Hosing, Regional, and Freshwater Fingerprints

Ryan Love[1], Lev Tarasov[2], Heather Andres[3], Alan Condron[4], Xu Zhang[5], and Gerrit Lohmann[6,7]

[1]Department of Earth Sciences, University of Ottawa, Ottawa, Ontario, Canada
[2]Department of Physics and Physical Oceanography, Memorial University of Newfoundland, St. John's, Newfoundland, Canada
[3]Physical Scientist, Oceanography, Northwest Atlantic Fisheries Centre, Fisheries and Oceans Canada
[4]Geology & Geophysics, Woods Hole Oceanographic Institution, Woods Hole, Massachusetts, USA
[5]State Key Laboratory of Tibetan Plateau Earth System, Resources and Environment (TPESRE), Chinese Academy of Sciences (CAS), Beijing 100101, China
[6]Alfred Wegener Institute, Helmholtz Centre for Polar and Marine Research, Bremerhaven, Germany
[7]Department of Environmental Physics & MARUM, University of Bremen, Bremen, Germany

**Correspondence:** Ryan Love (rlove@mun.ca)

**Abstract.** Freshwater, in the form of terrestrial runoff, is hypothesized to have played a critical role in past centennial to millennial scale climate variability by suppressing the production of deep-water in the North Atlantic. It may also play a central role in future climate change as ice sheet and glacier melt accelerates under anthropogenic climate change. In model studies of both past and future climate change, freshwater Hosing (i.e. injection across wide bands in the North Atlantic) is typically used

as a means-to-an-end by inducing a strong thermohaline circulation and climate response, with little regard for the other roles that freshwater plays in a complex coupled climate environment. Herein, we evaluate the realism of Hosing relative to two more sophisticated freshwater injection methods, all under glacial boundary conditions: regional injection (allowing the relatively coarse-resolution climate model to transport the freshwater) and a novel freshwater fingerprint method. The latter approach distributes freshwater into an eddy-parametrizing ocean model based upon where it is transported to in a higher resolution

eddy-permitting model. Here the COSMOS Earth system model is used as the eddy-parametrizing model, and a configuration of the MITgcm model as the eddy-permitting model.

This analysis address three primary questions. Firstly, where is freshwater routed at moderate versus eddy-permitting resolution? Secondly, does the freshwater fingerprint method allow the coarser resolution model to reproduce the net-results of eddy-permitting behaviour? Thirdly, how do climate impacts vary between different forms of freshwater injection?

Of the four outlets tested, we find that freshwater released at the Mackenzie River (MAK) outlet results in the most similar freshwater transport patterns at both the eddy-parametrizing and the eddy-permitting resolutions. However at eddy-permitting resolution there is a greater freshening of the contemporary Labrador Sea deep water formation region and mixing of freshwater into the Icelandic Sea. Similarly, Fennoscandian (FEN) freshwater discharge at eddy-permitting resolution results in more freshening of the Greenland-Iceland-Norwegian (GIN) Seas. Freshwater released from within the Gulf of St. Lawrence (GSL)

demonstrates large-scale differences between coarse-resolution versus eddy-permitting distributions over the whole North Atlantic. At eddy-permitting resolution, GSL-sourced freshwater freshens the North Atlantic (specifically deep water formation





regions important during cold glacial periods) more effectively than at eddy-parametrizing resolution. In COSMOS, freshwater deposited into the GOM results in a larger salinity anomaly at sites of deep-water formation in the northern North Atlantic than freshwater deposited into the GSL, whereas this relationship is reversed in MITgcm. When instead the freshwater fingerprint method is used in COSMOS, the relative strengths of GOM and GSL salinity anomalies in this deep-water formation region align with the eddy-permitting responses. However, for all injection locations considered, the pattern of MITgcm salinity anomalies are best matched by the COSMOS regional injection simulations and not the fingerprint injection simulations.

Hosing the North Atlantic in COSMOS results in stronger and earlier sea ice growth and surface cooling versus either regional injection or the fingerprint methods. Comparing regional injections to their respective fingerprint counterparts, injecting freshwater at the outlets results in earlier, but not faster, climate changes in the GIN Seas, mainland Europe, and GRIP ice core regions for the MAK and FEN regions relative to the fingerprint methods. For these injections, regional simulations show both a stronger and faster initial AMOC reduction than the fingerprint simulations. In short, for at least the COSMOS model, both the hosing and fingerprint methodologies are inferior to regional injection when considering salinity distributions alone but when considering AMOC and freshening of deep-water formation regions the fingerprint method corrects the response of the GSL and GOM outlets.

# 1 Introduction

The paleoclimate archive indicates that glacial runoff entering the oceans as freshwater can have a profound impact on global climate through perturbations to the Atlantic Meridional Overturning Circulation (AMOC) and associated northward heat transport. Such perturbations can induce hemisphere-wide cooling on the order of decades to centuries, e.g. the Younger Dryas period (Teller et al., 2002) and the 8.2 ka cooling event (Hoffman et al., 2012). A key uncertainty in the understanding of the effect of freshwater injection into the ocean arises from the fact that most models used to examine the impacts are of insufficient resolution to resolve the eddy-scale features that affect transport and mixing. This is particularly true for paleoclimate studies, for which simulations of millennial duration are commonly required. If the freshwater is not distributed correctly, it is unclear what inferential value the simulations have.

This study assesses the impacts of three different methods of imposing freshwater in the ocean component of a Earth System Model: broad-band Hosing, regional outlet-specific injection, and a freshwater fingerprint. This latter method is a novel technique introduced in this study. Our approach is to examine the climatic differences between these different representations of glacial runoff, and highlight some of the direct climatological effects associated with the introduction of freshwater at various outlets around the Arctic and Atlantic Oceans.

Surface runoff, glacially-sourced or otherwise, is understood to impact deep-water formation by reducing salinity in regions of deep-water formation that stabilizes the water column, results in a buoyant freshwater 'cap', and facilitates sea ice expansion. The ensuing reduction in vertical convection decreases the generation of deep-water, which results in a slowdown of AMOC, further expansion of sea ice, and a resultant decrease in northward surface heat transport. In climate model simulations, this reduction of heat transport results in abrupt ($\mathbf{O}$(decades)) Northern Hemisphere-wide cooling. This timescale is in



agreement with inferences from various proxies that also show equally abrupt shifts in climate occurred during the last glacial, most notably the **GR**eenland **I**ce core **P**roject (GRIP) deep ice core record (see Kindler et al., 2014, for a recent temperature reconstruction).

The role of freshwater in simulated coupled climate systems has been explored by a relatively large number of studies (some foundational examples are Stocker et al., 1992; Manabe and Stouffer, 1997; Stouffer et al., 2006), with most focusing

on its effect on the AMOC. Freshwater forcing is sometimes used as a means-to-an-end to generate an AMOC response by releasing the freshwater directly into a band in the North Atlantic and bypassing its transport from continental sources [1]. Such investigations, therefore, ignore relevant freshwater/ocean interactions between the physical point of injection and the Hosing region. As well, in some studies, Hosing has otherwise been justified to compensate for ocean model inadequacies in the routing of either glacial runoff (e.g. Arctic-derived freshwater as in Peltier et al. (2006) and generalized ice-sheet runoff as in He, 2011),

or freshwater sourced from iceberg armadas such as during Heinrich Events (i.e. the Ruddiman Belt as in TRACE21ka, He, 2011). This justification for hosing is argued in part due to the inability of moderate-resolution ocean models, such as are commonly used in Earth System Models, to resolve the small-scale features (e.g. boundary currents and mesoscale eddies) that are important to the transport and mixing of freshwater. More often than not, hosing is presented *a priori* as a valid methodological choice with no additional justification (e.g. Lohmann et al., 2016; Brown and Galbraith, 2016; Jackson and

Wood, 2017; Zhang and Prange, 2020; Sherriff-Tadano et al., 2021). However, the choice of hosing into the Ruddiman Belt is only reflective of the solid flux of the ice sheet and does not capture the interaction between solid and liquid freshwater fluxes from an ice sheet (Condron and Hill, 2021). As well, eddy-permitting modelling by Condron and Winsor (2012); Hill and Condron (2014); Love et al. (2021) have demonstrated that the traditional Hosing band is not reflective of where freshwater is actually transported when released from continental runoff outlets. The above raises the question: are the errors and biases

inherent in freshwater forcing methodologies understood well enough to make robust predictions of climate change in the near future or distant past, particularly in the case of idealized Hosing?

While the degree of mismatch between eddy-parameterized and eddy-permitting [2] freshwater routing has not yet been quantified explicitly, some inferences from previous injection studies can be made. Multiple outlets of glacial runoff have been previously identified and examined (e.g. Tarasov and Peltier, 2006). Of these outlets, the foci of this work are the Mackenzie

River (Northwest Territories, Canada/ Arctic Ocean), the Gulf of St. Lawrence (Quebec and Maritime provinces, Canada / North-East Atlantic Ocean ), the Mississippi River (Louisiana, USA / Gulf of Mexico), and Fennoscandia (North-West Atlantic Ocean). The location of each outlet is shown in Fig. 1. One of the more comprehensive studies examining the impact of injection into different outlets is Roche et al. (2009), whose coarse resolution (3° horizontal in the ocean, 120x65x20, rotated spherical grids, see  Goosse et al., 2010), eddy-parametrizing simulations are conducted using the LOVECLIM Earth system

Model of Intermediate Complexity (EMIC) under last glacial maximum (LGM) boundary conditions. Unfortunately, this study only presents results in terms of AMOC impact. As such, the magnitude of AMOC changes are interpreted as an indication

---

[1]This idealized form of freshwater forcing at any flux will be referred to as 'Hosing'.

[2]Here "eddy-permitting" indicates a horizontal resolution of at least one grid cell per Rossby radius and eddy parametrizing for coarser horizontal resolutions as per Nurser and Bacon (2014) .




of the amount of freshening at deep-water formation sites. Lohmann et al. (2020) provides a mix of eddy-parametrizing and eddy-permitting conditions through their use of an unstructured mesh approach with the FESOM model. In that study, eddies are generally resolved along coastal boundaries, but in the oceans' interior, including along the Gulf Stream, eddies are not explicitly resolved. Finally, Love et al. (2021) and Condron and Winsor (2012) both use a configuration of the Massachusetts Institute of Technology General Circulation Model (MITgcm) which is eddy-permitting for all regions equator-ward of $60°$ N. We note that no freshwater injection study to date (including the work presented here) is fully eddy-permitting in the Arctic, but the $\approx 18$ km horizontal resolution of the MITgcm (as used in Condron and Winsor, 2012; Love et al., 2021) is comparable to the Rossby radius determined by Nurser and Bacon (2014) in some Arctic regions (e.g. the Beaufort Gyre region).

All of these studies show consistency in the model response to injection of freshwater into the Mackenzie River and Fennoscandia. Differences arise when considering injection into either the Gulf of St. Lawrence or at the mouth of the Mississippi in the Gulf of Mexico. While the results of Roche et al. (2009) indicate that the Gulf of St. Lawrence injection is the least effective injection location at inhibiting NADW export relative to the outlets studied here, both Love et al. (2021) and Lohmann et al. (2020) show that such injection results in a strong freshening in North Atlantic sites of deep-water formation. As regards to injection into the Gulf of Mexico, again both Love et al. (2021) and Lohmann et al. (2020) are in agreement with freshwater from the Mississippi resulting in the weakest reduction of AMOC. By comparison, AMOC reduction for freshwater sourced from the Gulf of Mexico in Roche et al. (2009) is comparable to that sourced from the Labrador Sea and stronger than the Gulf of St. Lawrence. In summary, these results suggest that at least for locations near the Arctic, eddy-parametrizing model response to regional freshwater injection is consistent with that of high-resolution eddy-permitting simulations, but this is not the case for injection locations further south. However, in none of these cases does it appear that direct Hosing is an appropriate replacement. It is one goal of this study to elucidate the reasons behind these results by addressing this sort of mismatch explicitly through direct comparisons.

Given that eddy parametrizing models cannot accurately simulate the pathway of coastally released meltwater, we ask whether introducing freshwater into these models based on a fingerprinting technique (i.e. releasing it in locations that eddy-permitting models suggest freshwater would have been transported) offers an improvement. Herein, we present and test a freshwater fingerprint approach to mitigate the coarse resolution of ocean models in a resource-efficient manner. The fingerprint is a freshwater injection distribution generated from a model capable of resolving the $O(< 50$ km$)$ features of surface ocean circulation. These distributions, such as those shown in Fig. 1, spatially weight freshwater injection fluxes in coarse resolution models to correspond to the distribution that was obtained with eddy-permitting modelling. Another goal of this study to investigate the extent to which the climate response to freshwater injection via hosing differs from more realistic injection methods, as well as the impact of the specific geographical location of the injection on the climate system.

The goals of this investigation can be summarized by the following: Firstly, where is freshwater routed at moderate versus eddy-permitting resolution? Secondly, does the freshwater fingerprint method allow the coarser resolution model to reproduce the net-results of eddy-permitting behaviour? Finally, how do climate impacts vary between different forms of freshwater injection?





**Figure 1.** Each of the freshwater fingerprints generated from MITgcm output and subsequently used in this study shown on the GR30 grid used in COSMOS. The fingerprints are derived from the average, over the last 5 simulation years, negative salinity anomaly for a continual freshwater injection of 2 dSv (0.2 Sv) from their respective outlets (outlets shown with red circles) in the MITgcm simulations. The fingerprints shown are conservatively remapped to the GR30 grid from the CS510 grid of the MITgcm. To obtain an injection rate in $\frac{m}{s}$ for a given fingerprint simply multiply the shown distribution by the desired volume flux in $\frac{m^3}{s}$. Fingerprints shown at $\frac{1}{6}^\circ$ resolution comparable to the native MITgcm resolution are shown in Fig. S1. CBS and OBS stand for Closed Bering Strait and Open Bering Strait respectively. MAK, FEN, GSL, and GOM represent Mackenzie River, Fennoscandia, Gulf of St. Lawrence, and Gulf of Mexico outlets respectively, as described in Table 1. The time intervals for the temporal averaging for these fingerprints are years 13-18, 15-20, 15-20, 11-16, for MAK, FEN, GSL, and GOM respectively.



## 2 Experimental Design & Methods

Two models of different complexity and spatial resolution are used to study the transport of freshwater in the ocean and the impact of this freshwater on climate, the MITgcm and COSMOS. MITgcm was configured globally at an eddy-permitting resolution ($\approx \frac{1}{6}^{\circ}$) to study freshwater transport from coastal source locations and to generate spatial freshwater 'fingerprint'

maps. COSMOS (a CMIP3-era, fully coupled atmosphere/ocean/sea-ice/land Earth System Model) is run at a coarser eddy-parametrizing resolution ($\approx 3^{\circ}$) and used to assess the different climate impacts of Hosing, regional freshwater release, and freshwater 'fingerprinting' on the sensitivity of the climate system to glacial meltwater forcing.

Additional details about the various components of the methodology are provided below, starting with a description of the eddy-permitting simulations performed using the MITgcm in section 2.1. The subsequent processing to derive the fingerprints

from the source salinity anomaly fields is expanded upon in section 2.2. Section 2.3 describes the COSMOS model configuration used for all the freshwater forcing simulations. Finally, the simulations are listed in Table 1 and are described in more detail in Section 2.4.

### 2.1 Fingerprint Source Data

All of the fingerprint generation simulations are performed using the MITgcm using hybrid Last Glacial Maximum (LGM)/Younger-

Drays(YD) background conditions. The coupled sea-ice/ocean model is run in a Cubed-Sphere 6x510x510x50 (CS510) configuration, which provides an $\approx 18$ km horizontal resolution globally with 50 vertical levels. This grid geometry is eddy-permitting for most of the oceans (Chelton et al., 1998) with the exception of the Arctic (Nurser and Bacon, 2014). It is able to capture small-scale phenomena like coastal boundary currents, which are important for transporting freshwater runoff from the land in the ocean, more accurately than coarser resolution models such as those used in current and pre-

vious PMIP and CMIP working groups (Yang, 2003). For the LGM/YD hybrid background climate, sea level is set to that provided by the sea level solver component of the Glacial Systems Model of Tarasov et al. (2012) (GLAC1-D variant GLACDN9894GE90227A6005GGrBgic) at approx 13 ka, except for certain cases as regards the state of the Bering Strait. For all injection simulations except that from the Gulf of Mexico/Mississippi River (the one least likely to be affected by this choice), the Bering Strait is closed which is consistent with GLAC1-D. Fennoscandian and Gulf of St. Lawrence injection

simulations using the OBS configuration and associated fingerprints were also generated and are discussed further in Section S5.

The MITgcm was run in an ocean & sea-ice configuration with prescribed, monthly-mean surface forcings, cycled annually. Surface forcings are derived from a CCSM3 LGM simulation and have limitations with regards to overly zonal winds over the North Atlantic. This feature results in an overly zonal Gulf Stream with limited impacts on the results as discussed in Section

S2.

Several freshwater injection scenarios are examined using the following outlets: the Mackenzie River (MAK), the Gulf of St. Lawrence (GSL), the Gulf of Mexico (GOM)/Mississippi River, and a Fennoscandian (FEN) ice sheet adjacent outlet. A control simulation is conducted in parallel in order to determine the salinity anomaly distributions. These outlets are shown





| Run Short Name | Model | Continual Freshwater Forcing Distribution | Background Climate |
|---|---|---|---|
| OBS-Ctrl-MIT | MITgcm | None | OBS Hybrid LGM/YD |
| CBS-Ctrl-MIT | MITgcm | None | CBS Hybrid LGM/YD |
| MAK-MIT | MITgcm | 2 dSv MAK Regional | CBS Hybrid LGM/YD |
| FEN-MIT | MITgcm | 2 dSv FEN Regional | CBS Hybrid LGM/YD |
| GSL-MIT | MITgcm | 2 dSv GSL Regional | CBS Hybrid LGM/YD |
| GOM-MIT | MITgcm | 2 dSv GOM Regional | OBS Hybrid LGM/YD |
| Control | COSMOS | None | 38ka Orbital and GHG |
| Hosing-2d | COSMOS | 50-70N 2 dSv | 38ka Orbital and GHG |
| Hosing-1d | COSMOS | 50-70N 1 dSv | 38ka Orbital and GHG |
| Hosing-05d | COSMOS | 50-70N 0.5 dSv | 38ka Orbital and GHG |
| MAK-R | COSMOS | 2 dSv MAK Regional | 38ka Orbital and GHG |
| FEN-R | COSMOS | 2 dSv FEN Regional | 38ka Orbital and GHG |
| GSL-R | COSMOS | 2 dSv GSL Regional | 38ka Orbital and GHG |
| GOM-R | COSMOS | 2 dSv GOM Regional | 38ka Orbital and GHG |
| MAK-FP | COSMOS | 2 dSv MAK Fingerprint | 38ka Orbital and GHG |
| FEN-FP | COSMOS | 2 dSv FEN Fingerprint | 38ka Orbital and GHG |
| GSL-FP | COSMOS | 2 dSv GSL Fingerprint | 38ka Orbital and GHG |
| GOM-FP | COSMOS | 2 dSv GOM Fingerprint | 38ka Orbital and GHG |

**Table 1.** Table of both MITgcm and COSMOS simulations. The background hybrid LGM/Younger Dryas (YD) climate is as follows: 13 ka Bathymetry, LGM Ocean Surface Forcing, and a closed Bering Strait (with the exception of the GOM configuration which used a open Bering Strait, this is discussed in Section S5). The 38ka orbital and Greenhouse Gas (GHG) values are as follows: 0.013676 eccentricity, $23.2675°$ obliquity, $25.94°$longitude of perihelion, 185 PPMV $CO_2$, 405 PPBV $CH_4$, and 207.5 PPBV $N_2O$. The MITgcm simulations are comparable to those described in Love et al. (2021). Non-eustatic relative sea level changes are implemented with respect to the bathymetry in COSMOS, and the Bering Strait is closed in this configuration. We use the more appropriately scaled unit of dSv, 1/10 of a Sverdrup ($1 \, \text{Sv} = 1 \times 10^6 \, \text{m}^3\text{s}^{-1}$), to better reflect the magnitude of realistic freshwater fluxes.





with a red circle in Fig. 1. Each of these fingerprint source simulations are referenced by the location abbreviation and -MIT
(e.g. the Mackenzie River injection conducted with the MITgcm is referred to as MAK-MIT) as in Table 1. The freshwater
fluxes are 2 dSv imposed continually for the duration of the simulation. These freshwater fluxes are considered an analogue
for the outflow of solid and liquid mass from northern hemisphere ice sheets. Each simulation is $\approx 22-24$ years in duration,
after a 10 year-control simulation, which is of sufficient duration to resolve surface ocean transports of the freshwater signal
(Le Corre et al., 2020). Computation constraints preclude longer simulations with this model. We note that 2 dSv of freshwater
is an overestimate for FEN (Tarasov et al., 2014), but have elected to use this same value to allow for ease of comparison.
This hybrid boundary condition was used for several reasons. These runs were conducted as part of a prior investigation, Love
et al. (2021), and were readily available for this application. As well, the YD interval has more more robust constraints on
deglacial ice extent, topography, and land-sea mask for the YD interval relative to the Marine Isotope Stage (MIS) 3 interval.
Given the broad compatibility between the YD and MIS3 intervals, and the associated computational expense in repeating
these experiments for entirely consistent (and poorly constrained) MIS3 background conditions, this choice is a reasonable
compromise for a proof-of-concept investigation.

## 2.2  Fingerprint Processing

The fingerprints are derived from the salinity field difference between the injection and the control simulations. However,
to minimize the distorting effects of climate noise, differences of magnitude $< 3\sigma$ are set to zero. Here, $\sigma$ is the standard
deviation of daily mean salinities with respect to the multi-year monthly mean in the control run. While the assumption of a
Gaussian distribution for $\sigma$ is a simplification, an analysis of the distribution of salinity (not shown) reveals it is a reasonable
approximation. To mitigate any trends due to the configuration (i.e. model drift), the salinity fields are linearly de-trended prior
to calculating the control run $\sigma$ field. The difference between the injection simulation and its corresponding control is then
averaged over 5 years of each model simulation. This time interval provides a compromise between eliminating background
variability and obtaining a stable anomaly signal. In calculating the salinity anomaly, only those values which are reflective of
freshening are considered. The few features which have positive salinity anomalies are neither located near the injection region
nor large in magnitude, indicating that they are spurious noise. As such, these features are not considered in the fingerprint
generation.

At the next step in the procedure, the anomaly is vertically integrated over the whole water column to produce a 2D salinity
distribution. This integration has minimal effect on vertical salinity distributions in COSMOS, since the vast majority of the
anomaly is within the mixed layer of MITgcm and ends up being mixed vertically in the corresponding layer in COSMOS.
This distribution is area-weighted, then normalized to produce the fingerprint field. The fingerprints are provided at a global
1/6 degree resolution on a regular lat-lon grid. Figures S1 and 1 provide a comparison of the 1/6 degree grid relative to the
GR30 grid used by MPIOM (the ocean component of COSMOS with a grid size of 120x101x40). The normalized fingerprint
is conservatively remapped (i.e. the integral of the source field is preserved) to the GR30 grid using the first-order conservative
re-gridding method of the Climate Data Operators (CDO) toolset (Schulzweida, 2019) and implemented as a weighted injection
distribution into COSMOS simulations.



## 2.3 COSMOS Configuration

The COSMOS model is used for testing the climate impact of the different freshwater injection methods (Hosing, regional-injection, and fingerprint injection) discussed in Section 2.4, This model includes the the Max Planck Institute Ocean Model (MPIOM), the ECMWF Hamburg Version 5 (ECHAM5) atmosphere model, and the JSBACH land surface model. The atmospheric resolution is T31 ($\approx 3.75°$), with the land surface model sharing the same Gaussian grid. The ocean grid is nominally 3 degrees (grid size of 120x101x40 resulting $\approx 3°$ at the equator), with a displaced pole over Greenland and the other at the geographic South Pole, and thus has enhanced resolution for the Arctic, Labrador Sea, and Greenland-Iceland-Norwegian (GIN) Seas. The enhanced northern horizontal resolution results in average and minimum horizontal resolutions of $\approx 160$ km and $\approx 20$ km, respectively, versus the $\approx 270$ km global average. MPIOM is a primitive equation ocean model using the Boussinesq and hydrostatic approximations and uses a free-surface formulation (Jungclaus et al., 2006).

The model parameters are configured for 38 ka. We, as well as Zhang et al. (2021) who use a different ice sheet configuration, find this yields Dansgaard-Oeschger-like centennial to millennial scale internal climate variability in COSMOS. The 38 ka land ice, topography, and sea level configuration is provided via the methods and work presented in Tarasov et al. (2012), with associated updates where new glacial geological data (Carlson et al., 2018) has allowed for refinement of the GLAC1-D reconstruction. Non-eustatic sea level adjustments have been included in the configuration specifications. These non-eustatic adjustments are obtained via the gravitationally self-consistent sea level and glacial isostatic adjustment sub-models incorporated into the GLAC1-D reconstruction. This results in a thicker seawater column adjacent to ice sheets (i.e. near field regions) and a thinner seawater column for some regions distant from the ice sheet (i.e. far field regions). The Bering Strait and Canadian Arctic Archipelago are closed, and parts of the Arctic are glaciated further offshore than present land-sea boundaries. However, the Barents-Kara Sea remains open. These features result in a more isolated Arctic ocean by comparison to present day, but deep convection occurs in the GIN seas as in present day. Land surface types are as in COSMOS's pre-industrial datasets, but the plant functional types have been allowed to adjust to glacial conditions through the dynamic vegetation component of JSBACH.

As with the MITgcm simulations, all COSMOS injection simulations are compared to a control simulation (COSMOS-Control). This control simulation was branched from a previous MIS3 simulation with slightly different atmospheric and orbital forcings. The control simulation used here has been integrated forward for roughly 1000 years under the current climate conditions prior to initializing the injection simulations. The control and injection simulations all feature the following orbital and greenhouse gas forcings: 38 ka orbital configuration (0.013676 eccentricity, $23.2675°$ obliquity, $25.94°$ longitude of perihelion), 185PPMV $CO_2$, 405PPBV $CH_4$, and 207.5PPMV $N_2O$.

Note that the boundary conditions specified by the land-ice/sea level reconstruction for 38 ka are similar to the Younger Dryas configuration that was used to generate the fingerprints in regards to the sea level, ice sheet volume, and ice area. See Fig. S2 for the two paleoclimate land-sea masks and Fig. S3 for elevation and sea level differences between the time slices. Furthermore, both MIS3, which encompasses several DO events, and the Bølling-Allerød/Younger-Dryas are periods with centennial-millennial scale variability between warm interstadial and cold stadial intervals.




## 2.4 Freshwater Forcing Experiments

The configuration described in section 2.3 is used to test the relative climate impact of three methods of continual freshwater injection of varying complexity. These methods are, in order of increasing complexity, conventional Hosing, regional freshwater

injection (i.e. injection at the outlets and freshwater transported internally by COSMOS, the eddy-parametrizing model), and freshwater fingerprint (as described in section 2.2). The injection scenarios are as listed in Table 1. Freshwater in the 'Hosing' simulation is distributed continuously and uniformly across the 50-70N band in the North Atlantic at a rate of 2 dSv. This type of forcing is typical for a scenario where the goal of Hosing is a strong reduction of thermohaline circulation in the North Atlantic (e.g. Manabe and Stouffer, 1995), when testing climate sensitivity (e.g. Swingedouw et al., 2009), or when loosely

approximating the Ruddiman Belt ($40 - 65°$ N as per Ruddiman, 1977). The regional injection allows for comparison of the freshwater distribution generated by the COSMOS model to that generated by the MITgcm, as illustrated by the fingerprint. The regional injection locations used in COSMOS are the same as with the MITgcm fingerprint generation simulations (i.e. MAK, FEN, GSL, and GOM) but conservatively re-gridded to the GR30 grid such that the model inputs are equivalent. Finally, the freshwater fingerprint injection is implemented by taking the spatially variable, normalized freshwater fingerprint on the GR30

grid (as discussed in section 2.2) and scaling it by the 2 dSv flux. As with the MITgcm simulations, the naming convention of simulations is comprised of the abbreviation for the outlet and either -R for regional injection simulations or -FP for fingerprint injection simulations. All freshwater forcing is introduced into the combined liquid precipitation and surface runoff flux field directly within the ocean model. At 2 dSv, this is equivalent to an $\approx 1.5\%$ increase in global precipitation. There is no direct modification of atmospheric components from the addition of this freshwater flux.

All forcing simulations are compared to the control simulation detailed in Section 2.3. The control simulation oscillates between a cold phase and a warm phase without an external driving forcing (e.g. transient freshwater fluxes as in Brown and Galbraith, 2016). To minimize the effect of these oscillations upon the results, the injections are begun after the warm mode has hit a relatively steady state. This steady state is several hundred years after a transition from a cold to a warm state and $400$ years before a return to cold conditions. The choice of injection during an interstadial (warm) state is motivated by the

expectation of enhanced glacial runoff compared to that of a stadial (cold) state.

## 3 Results and Discussion

The results of this study are organized according to the research question they address. Where is freshwater routed at moderate versus eddy-permitting resolution? Does the freshwater fingerprint method allow the coarser resolution model to reproduce the net-results of eddy-permitting behaviour? How do climate impacts vary between different forms of freshwater injection?

The results and discussion will focus largely on the early decades after the initialization of freshwater injection. The reasons for this are twofold. Firstly, this allows for comparisons between this work and other investigations conducted at eddy-permitting or eddy-resolving resolutions which are limited to shorter durations due to computational constraints (e.g. Condron and Winsor, 2012; Love et al., 2021). Secondly, all COSMOS simulations eventually reach a stadial-like state where the extensive interstadial to stadial climate changes (surface air temperatures, AMOC, etc.) dwarf the more subtle freshwater injection





signal. The examination of the impact of freshwater injection on the characteristics of Dansgaard-Oeschger-like variability is left to future work.

### 3.1 Where is Freshwater Routed at Moderate Resolution, and Is Hosing Justified?

To facilitate graphical comparison, the COSMOS regional injection salinity distributions are processed using the same methodology and time-intervals as was used to generate the MITgcm fingerprints, creating COSMOS fingerprints. Comparing these
COSMOS-derived fingerprint fields to the MITgcm fingerprints, Figs. 1 and 2 respectively, some notable differences arise. As well, vertically averaged salinity anomaly plots for the upper 30 m and $\approx 100$ m for the COSMOS-R and MITgcm simulations over the same time intervals used in the generation of the fingerprints are shown in Figs. S4 and S5. A slower rate of transport at coarser resolutions is apparent when comparing the fingerprints and the salinity fields. There is less accumulation of freshwater in marginal seas along the pathway of freshwater (e.g. Baffin Bay, mostly impacts MAK-R) at eddy-parametrized resolution.
As well, there is greater freshwater transport across the Gulf Stream from both GSL-R and GOM-R at eddy-parametrized resolution. Each of these differences are discussed in more detail below. As well, marginal gradients in the eddy-permitting configuration are also generally sharper when compared to the coarse resolution configuration. This feature highlights the strongly diffusive nature of the eddy parametrizing configuration, a feature that has been shown to lead to unrealistic freshwater transport pathways in eddy parametrizing models (Condron and Winsor, 2012). Further discussion of this feature is found
in Section S3.

Previous studies indicate that passive tracers tend to be transported more slowly at lower resolutions than at higher resolutions ($1°$ and $0.1°$ respectively in Weijer et al., 2012). A similar difference is found here with respect to the rate of freshwater transport when examining the distributions in Figs. 1 and 2. Freshwater distributions averaged over years 10–20 of the regional injection simulations are shown in Fig. 3 where salinity anomalies are averaged vertically over the top 30m. MAK-R and FEN-R (Fig.
3E and J) both result in freshwater remaining mostly in the Arctic Ocean, with lesser amounts exiting via the East Greenland current than in the MAK-MIT and FEN-MIT simulations. Within the Arctic Ocean, the MAK-R freshwater spreads both east and west of the injection location, spreading as far west as the East Siberian Sea. In contrast, the FEN-R freshwater mass fills the central Arctic comparatively much less, with a greater proportion circling around the GIN Seas instead. Within the GIN seas, MAK-R freshwater appears to be fully entrained in the EGC, while MAK-MIT show freshwater penetration into the
Icelandic Sea north of Iceland.

Marginal seas are freshened less effectively in the eddy parametrizing configuration due to the aforementioned time lag and additionally, the small size of the marginal sea entrances relative to the model resolution. Baffin Bay experiences about half the freshening effect (relative to the total freshwater anomaly as in the fingerprints) in the MAK-R (Fig. 2) simulation as in the MAK-MIT (Fig. 1) simulation over the same time interval. Similarly, GOM-R and GSL-R show reduced (or no) freshening of
the Mediterranean Sea relative to their eddy-permitting MITgcm counterparts. For Baffin Bay an additional decade (see Fig. S6) is required in the MAK-R simulation before achieving comparable freshening to MAK-MIT. By comparison, GSL-R takes an additional $\approx 70$ years to freshen the Mediterranean Sea comparably to the levels seen in GSL-MIT. Thus, the net result is a less fresh Mediterranean Sea and Baffin Bay after a given period of injection for comparable injection volumes at lower





**Figure 2.** Freshwater distribution in the COSMOS simulations with specified regional injection using a comparable time range as their corresponding MITGCM fingerprint simulation (for Mackenzie River the CBS fingerprint time range was used rather than the OBS). The distribution was calculated using the same methods as to generate the corresponding MITGCM fingerprint. As in Fig. 1, the time intervals for the temporal averaging for these fingerprints are years 13-18, 15-20, 15-20, 11-16, for MAK, FEN, GSL, and GOM respectively.



**Figure 3.** Salinity anomaly in the COSMOS simulations time averaged over years 10–20 of injection vertically averaged over the top 30 m of the water column. Subfigures a,b,c show the impact of a 50-70N injection, region highlighted in yellow in figure S2, with varying fluxes uniformly distributed over the band. Subfigures d,f,g,h, show the salinity anomalies resulting from using the fingerprint distributions shown in figure 1 with a 2 dSv flux. Similarly, subfigures e,j,k,l show the salinity anomalies resulting from the regional injection locations as shown in fig S2. All runs shown are for the CBS configuration.





resolutions. For Baffin Bay, this has implications for rapid climate change, as this region can act as a reservoir of low-salinity
water that could then rapidly spread into the North Atlantic should some event result in a flushing action (e.g. deglaciation of
the Canadian Arctic Archipelago or Nares Strait).

Similarly, more freshwater is built up at the separation point of the Gulf Stream and North America in GSL-MIT than GSL-
R. This may be a result of differences in the location of the separation point of the Gulf Stream from North America, which
is further south in the MITgcm simulations than in COSMOS. In the GSL-MIT simulation, the freshwater flows southwards
towards Cape Hatteras (blue circle in Fig. S2) before flowing eastward across the Atlantic. Conversely, in the GSL-R simulation
freshwater begins to cross the Atlantic via entrainment in the Gulf Stream as soon as it clears the Cabot Strait (purple circle in
Fig. S2).

The bulk of the freshwater mass in GSL-R is directed south of the Gulf Stream into the sub-Tropical gyre when it reaches
the coast of Europe. While this pathway is represented in the freshwater distributions of GSL-MIT, the bulk of the freshwater
mass is redirected northward instead. Freshwater in GOM-R more readily crosses the Gulf Stream than GOM-MIT along the
East Coast of North America but otherwise follows a similar pathway as in the MITgcm. That is, they both follow the coastal
boundary currents at the Cabot Strait. About midway across the Atlantic, the injected freshwater in the GOM-R simulation
crosses the Gulf Stream and begins flowing northward. In comparison, freshwater in the GOM-MIT simulation only crosses
the Gulf Stream at the western boundary of the Atlantic.

The excess northward transport across the Gulf Stream may be partly explained by the fact that the freshwater flows out
of the GOM into the sub-Tropical North Atlantic more readily due to the lack of Caribbean Islands. At the equator, the grid
resolution is too coarse to adequately resolve the Caribbean Islands without creating an artificial land-bridge to Central America
or closing the Florida Strait. The effect of this feature ought to be greater exchange between the Gulf of Mexico and the Atlantic
Ocean and thus more freshwater into the Atlantic. This feature is not unique to the configuration, at least both LOVECLIM
(as in Bahadory et al. (2021), T. Bahadory, 2021 P. Comm[3]) and PLASIM (as in Andres and Tarasov (2019), H. Andres, 2021
P. Comm [4]) feature default model configurations with a similar representation. The net result of these differences is that the
coarser-resolution configuration shows more rapid transport of freshwater to sites of deep-water formation from locations south
of the Gulf Stream by comparison to the eddy-permitting configuration.

Having explored the differences between the routing of freshwater at eddy-parametrizing and eddy-permitting ocean model
resolutions, the question of whether these differences are problematic enough to justify bypassing freshwater transport alto-
gether and replacing it with Hosing is evaluated. If Hosing is to be considered an acceptable simplification it needs to repro-
duce the major climatological features (particularly AMOC) of the eddy-permitting models better than the regional injection.
However, at present there are no readily accessible, fully-equilibrated, coupled climate model simulations which utilize an
eddy-permitting ocean model to use as a reference to test this simplification. As such, the results of the regional injections with
COSMOS, and work of Love et al. (2021) provide the best data for comparison relative to the impacts of Hosing. The AMOC
response in the simulations are shown in Fig. 4. A key feature is that Hosing produces a larger and more rapid change than any

---

[3]Confirmed lack of Caribbean Islands in LOVECLIM

[4]Confirmed lack of Caribbean Islands in PLASIM





**Figure 4.** AMOC (maximum of the streamfunction in the North Atlantic between $20 - 70°$ N and below 700 m) for each of the freshwater injection scenarios. In all panels the thick red line denotes the control simulation. a shows the Hosing scenarios for 3 different values, b shows the regional injections where the freshwater flux is introduced at the locations shown in figs. 1 and S2, and c shows the different fingerprint scenarios. All injections are conducted with 2 dSv injection volumes unless otherwise noted in the legend. Data shown has been processed using 10 year window running mean.





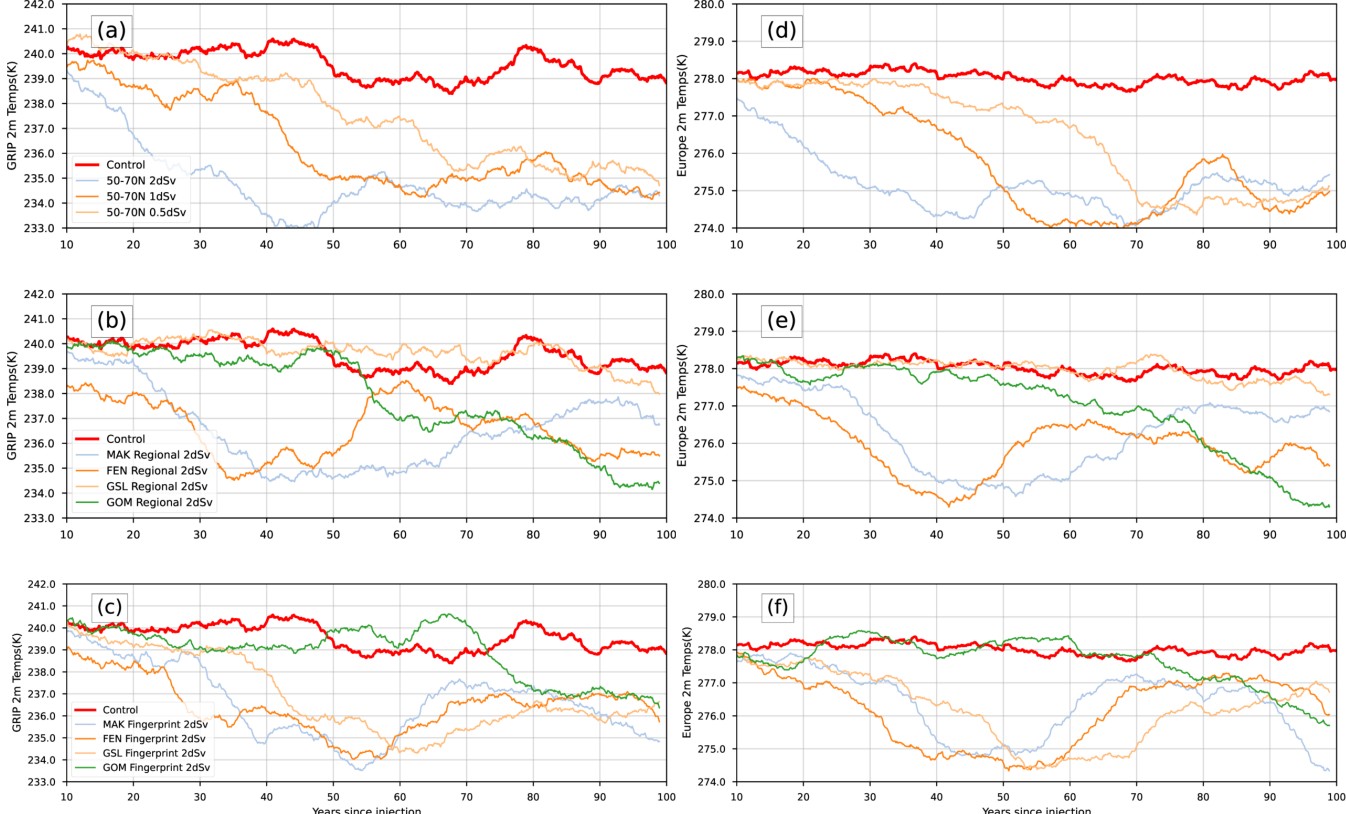

**Figure 5.** Surface temperatures at GRIP and continental Europe from each set of simulations. a and d show the Hosing scenarios, b and e show the regional injections as shown in Fig. S2, while c and f show the fingerprint injection scenarios as in Fig. 1. Data shown has been processed using 10 year window running mean.

regional injection considered here for the same flux of freshwater. For example, a 1 dSv hosing injection results in a greater peak reduction than MAK- or FEN-sourced freshwater at twice the injection rate.

Hosing also generally results in a stronger and faster cooling impact relative to regional injection, even when comparing
1 dSv of Hosing to 2 dSv of regional injection. Figs. 5A,B,C show that for the same flux of freshwater, the simulated, maximum GRIP 2m temperature deviation is at least 1 C colder than the MAK-R or FEN-R injections. Similarly for continental European 2m temperatures, Figs. 5d,e,f, demonstrate this same feature. The cooling is also spread over a much greater area, as seen in Fig. S7, where equivalent fluxes applied via Hosing generate cooling covering much of North America, Europe, and North Asia, instead of the more regional cooling of the regional injection scenarios. Sea ice cover growth demonstrates a faster and
stronger overall response to Hosing, relative to the regional injections, is shown in Figs. 6, S8, S9.

Thus, the deficiencies introduced by Hosing invalidate its use in comparison to regional injection. Nevertheless, there may be some circumstances in which Hosing is justified. One example may be climate sensitivity tests between models. Hosing may also be required in order to compare results against past work. One simple method to compensate for some of the defi-



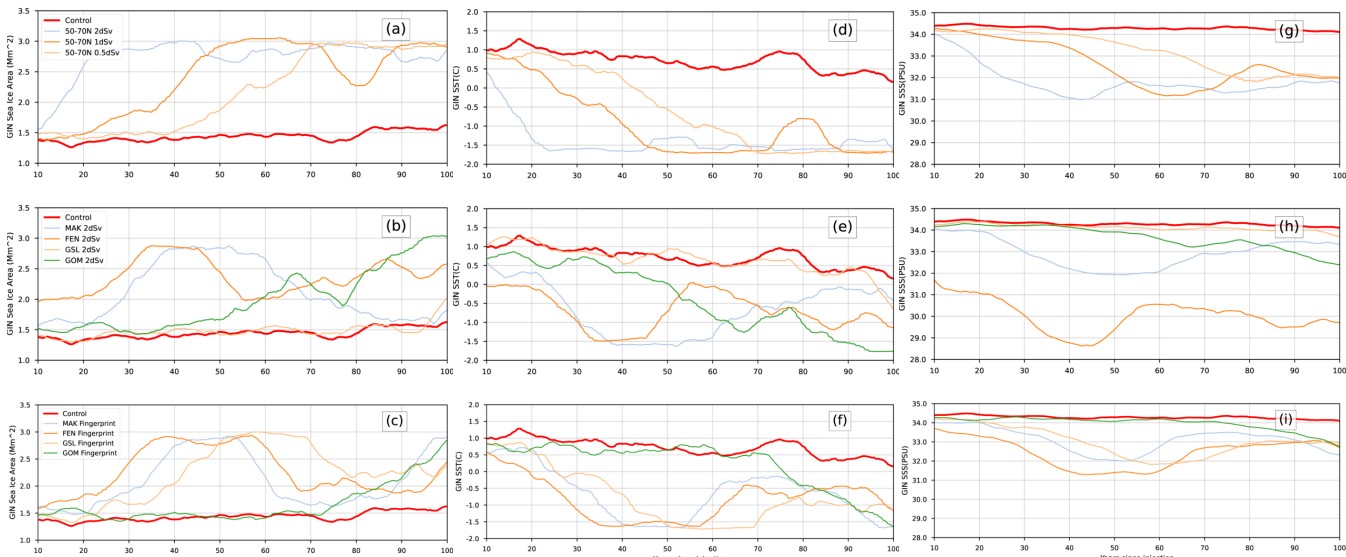

**Figure 6.** Ocean domain metrics for the upper layers of the Greenland-Iceland-Norwegian Seas for each set of injection scenarios. Panels a/b/c/, d/e/f, and g/h/i show sea ice area, sea surface temperature, and sea surface salinity respectively. Panels a/d/g/, b/e/h, and c/f/i are for the hosing, regional injection, and fingerprint injection methods respectively. Each of the time-series shown are boxcar running mean with a 10year window.

ciencies introduced by Hosing might be to reduce the effective freshwater flux. For example, reducing fluxes by a factor of 4
as per the findings from Love et al. (2021) would replace 2dSv of freshwater by 0.5 dSv which is shown in Figures 4, 5, and
6 to be a better representation of the climate effects from freshwater from some gateways at least. As a reminder, though, this
simple correction does not account for the non-linearities and differences in spatial distribution shown in Figure 3. The results
presented here show that progressing from Hosing to regional injection yields an increase in realism (i.e. upstream salinity
anomalies with associated impacts, less extreme climate responses) while still eliciting a strong AMOC response, and associ-
ated climate cooling. However, there remains the problems associated with regional injection at eddy-permitting resolutions
(e.g. that freshwater crosses the Gulf Stream in unrealistically high amounts). It is these problems that the Fingerprint method
attempts to address.

### 3.2   Does the Fingerprint Method Capture Eddy-Permitting Behaviour?

Given that the regional injection results, and those of previous studies, fail to adequately reproduce the salinity distributions
from an eddy-permitting configuration for freshwater sourced from the GOM and the GSL, an alternative method is required.
As such, the freshwater fingerprint technique as described in section 2.2 is examined. The salinity distributions in the fingerprint
forcing simulations shown in Fig. 3d,f,g,h, show some regional signature of the fingerprint forcing distributions themselves (as
shown in Fig. 1). However, the salinity distributions produced from the regional injections more closely resemble those of the



eddy-permitting simulations (as seen in Figs. S5 and S4) when considering equivalent time-intervals. This is not unexpected
given the fingerprint approach effectively introduces the glacial runoff/freshwater downstream and does not presently account
for existing freshwater transports in COSMOS.

Examining the difference in the salinity anomalies between regional injection and the fingerprint injection method in COS-
MOS, shown in Fig. S10, and the resulting climate impacts, several differences are notable. The maximum freshening of the
GIN seas is reduced with the fingerprint method (see as well Fig. 6), likely due to the more diffuse nature of the fingerprint
injection method. Much like the MAK-MIT injection simulations, the bulk of the freshwater in the MAK-FP simulation is
transported from the Arctic to the North Atlantic via the East Greenland Current and the Labrador Sea. However, one of the
main features of both the MITgcm and regional injection scenarios not reproduced using the fingerprint method is the central
core of freshwater built up around the site of the freshwater injection. Instead, the fingerprint by design emulates one of the
main functions of this water mass (providing a reservoir for downstream freshwater flow), by virtue of sourcing the freshwater
along the transport pathways from the total injection amount.

The second discrepancy between the COSMOS regional injections and the MITgcm transports discussed in section 3.1
is the lack of freshwater build-up in marginal seas. The GSL-FP simulation exhibits a freshening of the Mediterranean Sea
comparable with the GSL-MIT simulation, but with no appreciable overshoot. This discrepancy for freshening of marginal
seas demonstrates the limitations of comparing model results for these regions to proxy evidence. Furthermore, the previously
discussed flushing behaviour of a region like Baffin Bay would be very sensitive to the fingerprint. For example, if exploring
the concept of Baffin Bay as a reservoir of lower salinity water, the use of the MAK or FEN fingerprints would enhance the
impact of flushing due to increased freshwater input to Baffin Bay relative to the other fingerprints.

Finally, one of the most important contrasts between the regional injections and their MITgcm counterparts is the rate
of freshwater transport across the Gulf Stream. As discussed in section 3.1, GSL-R's freshwater is largely entrained in the
sub-Tropical gyre after crossing the North Atlantic, whereas GSL-MIT's freshwater is largely directed northward to the deep-
water formation (DWF) site. When using the fingerprint injection method for this outlet, the balance shifts and there is a greater
amount of freshwater in the region of DWF. Examining the impact of this shift on AMOC in Fig. 4, it is readily apparent that this
shift results in the GSL-FP having a more rapid effect on reducing AMOC. In comparison, using the GOM fingerprint results
in a lesser rate of AMOC reduction by comparison to the regional injection. As discussed previously, GOM-R's freshwater
crosses the Gulf Stream midway across the North Atlantic. Through this pathway, the freshwater begins to flood the DWF
region approximately 50 years after the start of injection. Freshwater in the GOM-FP simulation does not follow this pathway
and instead reaches the DWF region via the eastern edge of the Atlantic and not in as great a concentration as the regional
injection. These differences result in a delay in AMOC reduction and a lesser rate of decrease relative to the regional injection.

Due to the short integration time of the MITgcm simulations, to evaluate the impact of the fingerprint method in the eddy-
parametrizing simulations relative to the eddy-permitting simulations, the relative amounts of freshwater transported to sites
of deep-water formation in the eddy-permitting simulations need to be examined. In doing so, we find that for all outlets the
relative AMOC impact of the fingerprints injection methods more closely resemble the amounts of freshwater transported
to sites of deep-water formation in the eddy-permitting MITgcm simulations. That is, the use of the fingerprint reduced the



AMOC impact of the GOM injection and increased the impact of the GSL injection, such that the relative climate impacts

now reflect that of the eddy-permitting MITgcm simulations. This indicates that for at least the GSL and GOM outlets, the fingerprint injection has had the desired effect of letting us emulate at least one key climate impact, the AMOC.

In summary, when considering the influence of freshwater on AMOC alone, the fingerprints mitigate some issues for freshwater introduced into the GOM and GSL, specifically the greater impact of GSL sourced freshwater on AMOC. In spite of this result, the use of fingerprints results in a freshwater distribution more dis-similar to the eddy-permitting simulations than the

390 regional injections for all injection sites considered. As well, the fingerprints themselves are sensitive to climate mismatches, e.g. Gulf Stream location, between the coarse resolution simulations and the eddy-permitting source simulations. As the fingerprint injection doesn't account for existing transports in COSMOS, a possible solution is to apply the fingerprint to only some fraction of the discharge and have the rest discharged regionally. A brief examination of this approach is discussed in Section S6 for the MAK region. However, we find that such blending of fingerprint and regional injection masks does not significantly

improve the representation of the salinity anomaly by comparison to regional injection alone.

### 3.3 How do Climate Impacts Vary Between Different Forms of Freshwater Injection?

Here the associated climate impacts that arise from the differences between the different forms of freshwater injection are discussed. The discussion focuses on relative changes in the GIN Seas sea ice area (SIA), sea surface salinity (SSS) and sea surface temperature (SST). These features ought to change in parallel as they are tightly coupled. A fresher ocean or

400 colder ocean surface results in sea ice growth. However, sea ice growth results in brine rejection, resulting in a more saline environment. Sea ice also acts as a heat sink, keeping the surface at the freezing point until melted.

Generally, changes in these characteristics of the GIN Seas reflect similar changes in the North Atlantic and the Arctic. As such, these other regions are only discussed in detail where differences arise. As well, for this configuration of COSMOS we note deep convection occurs in the GIN seas. Time series of the data discussed are shown in Figures 6, S8, and S11 for the GIN

Seas, the North Atlantic, and the Arctic respectively. All GIN Sea metrics for the Hosing scenarios increasingly non-linearly trend away from the control scenario for ever-increasing rates of injection. Hosing is highly effective at freshening the surface layers of the GIN Seas and North Atlantic for both the 2 dSv and 1 dSv cases, as expected. The 0.5 dSv case does not show a freshening effect within the first 10 years, taking $\approx 30$ years of injection to have an overall negative salinity anomaly in the GIN Seas and North Atlantic, which correlates with the initial reduction in AMOC for that injection scenario. For 2 dSv of

injection GIN Seas SIA grows at a rate of approximately $0.1 \mathrm{Mm}^2/$ year, and both 1 dSv and 0.5 dSv grow at $0.05 \mathrm{Mm}^2/$ year. [5]. Terrestrial temperature time series in Fig. 5 reproduce the sea ice behaviour. Hosing results in a faster and more sustained change in SIA, SSS, and SST relative to regional or fingerprint injections. This difference is more dramatic when examining 2m temperatures with 2 dSv Hosing resulting in a peak cooling $> 2$ deg C greater than any other injection method. There are similar relationships between the FEN-R and MAK-R in the North Atlantic when comparing SIA, SSS, and SST changes

(see Fig. S8). Comparing these climate metrics and AMOC for GSL-R and GOM-R the same relative impacts are present: freshwater from GOM results in an overall greater climate impact than GSL.

---

[5]Rates calculated as a linear fit from zero sea ice change to peak sea ice coverage at $\approx 3 \mathrm{Mm}^2$



Comparing regional injections to their respective fingerprint counterparts, injecting freshwater at the outlets results in earlier, but not faster, SIA/SST/SSS changes in the GIN Seas region for the MAK and FEN regions relative to the fingerprint methods. This relationship is also present when examining 2m temperature changes for Europe and at the GRIP ice core in Greenland (see Fig. 5. This temporal relationship is at odds with the lag difference discussed in section 3.2, where the fingerprints demonstrated a faster response relative to the eddy-parametrizing simulations. As shown in Fig. 6, the rate of SIA changes for MAK-R relative to MAK-FP is higher, whereby sea ice fully covers the GIN Seas at roughly the same rate as FEN-R. This feature is due in part to the enhanced surface runoff from Eastern Europe/North Asia, as discussed in section S4. FEN-FP has similar results on GIN SIA and SSTs as FEN-R but these metrics remain at peak values for approximately twice as long a duration. Freshwater introduced in the GSL-R simulation does not result in any changes of note in the upper layers of the GIN Seas until close to the end of a century after the start of the injection, while the GSL-FP simulation begins freshening, cooling, and growing sea ice $\approx 25$ years after the start of injection. Both these features of the FEN-FP and GSL-FP simulations are present when using CBS or OBS derived fingerprints, as shown in Section S5.

GOM-R results in consistent growth of sea ice and freshening of the sea surface from $\approx 40$ years after the onset of injection despite being the furthest injection location from the GIN Seas. This is a result of the freshwater more effectively crossing the Gulf Stream as described in section 3.1. The GOM-FP compensates for this effect and is resultantly less effective than the GOM-R at freshening sites of DWF.

The AMOC response between the fingerprint simulations and the regional injection simulations for FEN and MAK are very similar. For these injections, regional simulations show both a stronger and faster initial AMOC reduction than the fingerprint simulations. This is due to the stronger salinity gradient and larger presence of freshwater at the surface in the regional simulations. The overall pattern of AMOC reduction for these simulations falls somewhere between the 2 dSv and 1 dSv Hosing simulations. All of these simulations eventually result in an AMOC strength of $\approx 12$ Sv at the end of a century of freshwater injection. Examining the impacts of using CBS vs. OBS fingerprints, as described in Section S5, we find no impact on our conclusions for the outlets investigated.

In summary, for traditional Hosing, results are obtained that are readily expected based on first principles and previous studies (i.e. strong cooling in the NH, severe AMOC reductions, etc.) For FEN-R and MAK-R, changes in relative sea ice area, SST, and SSS in the GIN Seas are comparable with the freshening effects observed in Love et al. (2021). However GOM-R shows more impact in the GIN Seas region than GSL-R, contrary to these previous results. This ordering is reversed and results align with Love et al. (2021) when using the freshwater fingerprints, demonstrating that the fingerprint method is able to emulate some of the behaviour present at the eddy-permitting scale in the coarse resolution configuration. The freshwater fingerprint runs show a delay in SIA, SSS, and SST in the GIN Seas and 2m temperatures over Greenland and Europe relative to their comparable regional injection simulations, due to the less concentrated injection regions



## 4 Conclusions

As motivated in the introduction, we have addressed the following questions: Where is freshwater routed at moderate reso-
lution? Does the fingerprint method reproduce the net-results of eddy-permitting behaviour? How do climate impacts vary
between different forms of freshwater injection?

We compare eddy-permitting simulations using the MITgcm and much coarser resolution ($\approx 3°$) eddy-parametrizing simu-
lations of the Max Planck Institute Ocean Model MPIOM integrated into COSMOS. Regional freshwater injection in the eddy-
parametrizing model can reproduce some, but not all, of the major features observed at the eddy-permitting scale. Regional
injection, for all outlets examined, results in slower freshwater transport and reduced marginal seas freshening at the coarser
eddy-parametrizing resolution. In particular for MAK, we find that the eddy-parametrizing configuration does not freshen the
Labrador Sea as effectively. This suggests that the impact on any deep water formation in that region would be reduced by
comparison to higher resolution configurations. Similarly, for the Fennoscandian (FEN) outlet we find the central GIN Seas
are freshened less effectively at the eddy-parametrizing resolution, an important consideration for model configurations with
deep water formation in that region.

These variations in marginal sea freshening between eddy-parametrizing and permitting scales will affect the ability of a
simulated marginal sea like Baffin Bay to act as a reservoir for low-salinity water. This could have important implications for
the simulation of rapid climate change. In particular, the lower salinity water of marginal seas at eddy-permitting resolutions
could rapidly spread into the North Atlantic if a flushing action were to take place (e.g. due to the deglaciation of the Canadian
Arctic Archipelago or the opening of Nares Strait, etc.).

The distributions of freshwater injected into the Gulf of St. Lawrence (GSL) and the Gulf of Mexico (GOM) are signifi-
cantly different across much of the mid-Latitude North Atlantic at eddy-parametrizing resolutions from their eddy-permitting
counterparts. This arises due to differences in the tendency of freshwater to cross the Gulf Stream at eddy-permitting and
eddy-parametrizing scales. Greater freshwater transport across the Gulf Stream at eddy-parametrizing resolution impacts the
transport to sites of deep-water formation from these locations, over-emphasizing the effect of freshwater from the GOM while
diminishing the impact of freshwater from the GSL. Despite these limitations, the results demonstrate that regional injection
for several key locations along the coasts of the Atlantic and Arctic oceans induce a strong AMOC response, and the associated
climate cooling, albeit not as swiftly or strongly as Hosing. Given the relative ease of implementing regional injection methods
and the overarching similarities in AMOC responses, we conclude that for the outlets investigated, Hosing is not an acceptable
simplification.

The fingerprint method, when compared against the eddy-permitting model results, alleviates some of the limitations of re-
gional injection (e.g. the aforementioned excessive transport of freshwater across the Gulf Stream). The freshening of marginal
seas is improved for both the Labrador and Mediterranean Seas when fingerprinting is used for MAK and GSL freshwater
injection, respectively. However, the overall salinity anomalies for the eddy-permitting model are more poorly captured by
simulations using fingerprint injection than by those using regional injection. The differences are largest at the location of the




source of glacial runoff. As such, the fingerprint method is of limited use when considering the impact of the salinity anomaly itself and as such is not successful in replicating the impact of eddy-permitting simulations.

An important caveat is that all these simulations neglect the role of bottom-riding (hyperpycnal) flow. Bottom-riding flow would impact the results for any outlet with sufficient sediment loading of the freshwater discharge for such flow to occur (e.g. GSL and GOM Broecker et al., 1989; Parsons et al., 2001; Tarasov and Peltier, 2005; Aharon, 2006). This phenomenon should result in a significantly reduced surface freshening signal, with a commensurate climate impact from modified NADW formation and AMOC. As such, the scale of the impact should be explicitly tested in a future investigation.

The different forms of freshwater injection considered here also impact climate differently. The overall climate response to Hosing is both faster and greater in magnitude compared to that of regional or fingerprint freshwater injection. When comparing the AMOC response, we find that GSL regional injection results in less of a response than the GOM injection. This result is in disagreement with deep-water formation region freshening results from the eddy-permitting simulations for these outlets. These same responses are inverted when using fingerprint injections, in correspondence with the eddy-permitting freshwater injection simulations. Hosing leads to an exaggeration of surface cooling in Greenland and Europe, as well as an exaggerated growth of northern hemisphere sea ice. As such, these biases need to be addressed when pursuing any investigations of climate stability or future climate projections if choosing to represent freshwater injection as a simple band over sites of deep-water formation. We have demonstrated that the climate system is sensitive to the representation of freshwater injection. Biases arising from the choice of injection method must therefore be accounted for in order to make robust climate predictions.

## 5 Acknowledgements

The authors would like to thank Dr. Guido Vettoretti for his assistance in reviewing this document as part of the thesis of RL. The authors would also like to thank those at the GNU and Fedora projects, Kernel.org and in particular those responsible for GNU Parallel (Tange, 2011) whose software greatly sped up and streamlined the analysis in this work. This is a contribution to the ArcTrain program, which was supported by the Natural Sciences and Engineering Research Council of Canada This work was supported by the PALMOD project. This research was also enabled in part by support provided by SciNet (www.scinethpc.ca) and Compute Canada (www.computecanada.ca) through both Resources for Research Groups allocations and the Rapid Access Service.

*Code availability.* The numerical models used in this research are available from their respective working groups.

*Data availability.* Select data will be made available in FAIR aligned repositories, with this section being updated as data is made available.



*Competing interests.* The authors declare no competing interests are present.

*Author contributions.* RL, HA, and LT prepared the experimental design. RL, HA, AC, and LT analysed the experiment results. AC provided
technical expertise for the MITgcm experiments. XZ and GL provided technical expertise with respect to the COSMOS experiments. RL
conducted the experiments and prepared the manuscript with contributions from all authors.



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
