# Peer review of "Exploring the climate system response to a range of freshwater representations: Hosing, Regional, and Freshwater Fingerprints"

_EGUsphere, 2023_

## Referee Comment (RC2)

**General comments**

This paper presents a large number of different simulations where freshwater is released in different locations in the North Atlantic – Arctic in glacial climate conditions for time scale lower than a century. It uses two different models: one OGCM at 1/6° of horizontal resolution and one AOGCM at 3° horizontal resolution in the ocean. In fact the OGCM is mainly used to derive a fingerprint of freshwater pathway that is then used within the AOGCM. A few differences concerning the sensitivity of the freshwater release is found and interpreted before to evaluate the climatic impact of the freshwater at the different locations.

The topic of the paper is of great interest given that there are now more and more studies highlighting the strong sensitivity on the way the freshwater released along the North Atlantic coast is spreading to the resolution of the ocean model used. This is crucial given that the meltwater from glaciers and ice sheets is actually largely released at the coast, while convection in the ocean generally occurs in the open ocean. Thus, depending on the lateral diffusion of the coastal freshwater signal towards the center of the gyres, the impacts of meltwater might be very different depending on the model used, and potentially not realistic at all in coarse resolution model like CMIP6 ones. Here the authors consider quite a high number of sensitivity experiments to tackle this question in the context of paleo timescale and glacial climate. The idea of fingerprinting the freshwater pathway using a high-resolution is interesting and might deserve to be published.

However, while this topic is highly relevant both for the past and the present, I find that the present study is not providing a very strong progress on the existing science. Indeed, the tools used are quite "old" (AOGCM from CMIP3 generation, while CMIP7 is now coming, and OGCM at 1/6° of resolution, which was already run more than 20 years ago, e.g. New et al. 2021 and the new standard for this kind of ocean-only model is rather around 1/20 to correctly solve the eddies, e.g. Hirschi et al. 2020 among many others). Thus, we cannot really say that the tools used are really state of the art. And this has important implications given that eddy rich models are really changing the dynamics of the AMOC, far more than eddy permitting (Hirschi et al. 2020) and are likely closer to the dynamics at play in the real ocean. This might be clearly stated!
Models with such coarse resolution are now still used since they can allow to perform long transient runs (multi-millennia) to analyse intrinsic AMOC variability during glacial time for instance, but here only simulations of 100 years or less are performed. Since this study aims at evaluating climate dynamics during paleo-periods, we can wonder if such short-time simulations are really relevant to improve our understanding of climate dynamics as compared to existing records. The implications of the results are also poorly discussed in regards of existing literature.

Last but not least, the paper suffers from other considerable issues like: not-in-depth analysis of the physical mechanisms at play, poor logical linkages in the text, lack of discussion of caveats and limitations of the study, and poor synthesis of the main results of the study. Thus, I support rejection and resubmission of a clearer and more in-depth analysis of the physics at play in their simulations.

**Specific comments**

- Abstract: It is quite long and not very clear. I agree that the experimental design used is quite complex and it took me time to understand it, but please try synthesize better what you've done and why (and leave details out of the abstract to focus on the main results).
- Line 68: you can consider to cite Swingedouw et al. (2022) to substantiate this claim (although this was using an eddy rich model).
- Line 75: the link with climate change is not obvious given that the focus here is on glacial time and paleostudies. Please state this more clearly.
- Line 88: unclear sentence? What was missing? Please be more explicit.
- Line 108: "cannot" is too strong. The parametrization aims at reproducing that. What you mean here is that they are not doing a good job at this, which I agree, but maybe with better parametrization this can be solved.
- Line 121-132: As far as I understand the use of MITgcm is only for finding the fingerprints that are then used in the coarse resolution OAGCM. This is not stated very clearly here, while the title of 2.1 states implies it but not that clearly. Please rephrase a bit to clarify this.
- Line 134-150: not much is said about the mean state of this MITgcm run. What is the strength of the AMOC, where are located the convection sites. Is the circulation realistic for glacial time?
- Line 165: replace "is" by "might be", since the use of only an eddy-permitting model is really questioning the realims of the results.
- Line 174: when are those 5 years selected (what time of the simulation).
- Line 181: this claim is not supported by any figures I think, so a "not shown" is necessary here.
- Line 198: if there are millennial intrinsic oscillation in COSMOS, how did you change the period of the control simulation? Is the AMOC changing on the long term?
- Line 210: since the focus here is on the freshwater release, the way this freshwater is accounted for in the model should be depicted. Is the model rigid lid or free surface? Which kind of parametrization? Is salt conserved with this parametrization?
- Line 213: still this question about intrinsic variability during glacial time?
- Line 249: within those questions, there is an implicit assumption that the response of model to freshwater only depends on their pathway, which is totally false, as shown in Stouffer et al. (2006) for the same design of freshwater release, the model responses can vary by an order of magnitude in terms of AMOC!
- Fig. 5: I do not find this analysis very enlightening: the curves are very messy and not much is said about the differences. Are they due to AMOC response? A correlation of the climate response this AMOC response (with dfferent lags) might be the least to be done. The residual might represent the effect of other processes than change in meridional heat transport due to the AMOC, including atmospheric noise, change in stratification, gye transport, etc.
- Line 389: unclear sentence.
- Line 404: this is "not shown", is it?
- Line 411; any correlation to support this claim?
- Line 417-419: unclear sentence.

- Line 435: replace "is" by "might be" since this is a hypothesis at this stage.
- Line 440: "first principles". What do you mean? Please be more specific here.
- Line 445-447: please rephrase, I do not get the reasoning here.
- Line 450-451: Fig. S7 should show significant results only! (using a t-test of the difference for instance).
- Line 465-475: since the site of convection are not shown (among many other things, e.g. barotropic streamfunction, etc.), I think that the analysis is in the end quite poor and not going much into depth of real understanding of the differences in response.
- Line 481-483: where is it shown?

**Bibliography**

Hirschi, J. J. -M., Barnier, B., Böning, C., Biastoch, A., Blaker, A. T., Coward, A., et al. (2020). The Atlantic meridional overturning circulation in high-resolution models. *J. Geophys. Res. Oceans* 125, e2019JC015522. doi: 10.1029/2019JC015522

New A. L. et al (2001) On the role of the Azores Current in the ventila- tion of the North Atlantic Ocean. Prog Oceanogr 48(2–3):163–194

Swingedouw D., Houssais M.-N., Herbaut C., Blaizot A.-C., Devilliers M. and Deshayes J. (2022) AMOC Recent and Future Trends: A Crucial Role for Oceanic Resolution and Greenland Melting? *Frontiers in climate*, 4:838310. doi: 10.3389/fclim.2022.838310.

---

## Author Comment (AC1)

**Response to Reviewer (RC1) :**

Firstly we would like to thank the anonymous reviewer for the time they have taken to read through the manuscript and make their suggestions. An itemized set of responses to their comments is below presented in the same order as the reviewer has provided.

*"The authors should synthesize all parts of the manuscript to make it easier to read. There are a lot of repetition throughout. The introduction needs to be re-structured to provide a more logical flow. At the moment, it seems like the introduction includes a summary and a longer description. In addition, the results are a bit difficult to follow. There are a lot of figures (or at least a lot of information in all the figures), and they are not necessarily mentioned when describing the results."*

We will re-evaluate the manuscript for the repetition and logical flow the reviewer comments on. We anticipate that addressing the specific comments both reviewers provide will be sufficient to address this point.

*"Please be more precise, particularly in the abstract and conclusions. For example, L. 28-35 and 488-498, the broad hosing is compared to the other methods in a very subjective way. The response is faster by how much? The climate response is greater by how much and when? ..."*

We will add further details regarding features (e.g., such as the speed of the climate response) to the conclusions of the manuscript and evaluate the suitability of such statements within the abstract (i.e., where brevity is of value).

*" - It would be nice to show a figure in the supplement of the surface currents under LGM/YD conditions and how they differ with the pre-industrial run. This could be compared to a similar figure done with COSMOS."*

We did not conduct pre-industrial control runs with either of the configurations in the manuscript. While we do see some utility for such a comparison, we no longer have the allocated compute resources with which to conduct such simulations. We will add plots of surface currents for both a MITgcm and COSMOS control simulation into the supplemental materials.

*"- Was salinity restoring used in the simulations performed with the MIT-GCM? If yes, how would that impact your results?"*

Salinity restoring was not used in any of these experiments. A line will be added to the experimental design section for clarity.

*"- I understand that if other modellers want to re-do similar experiments, then the injection rates as shown in Figure 1 are relevant, but I find the salinity anomalies much more useful and would suggest to show the top 30m salinity anomalies (top 100m for GOM?) as figure 1 instead."*

We disagree as the reviewer has misunderstood the contents of Figure 1, which is not rates but rather the salinity injection distribution derived from the salinity anomaly from the MITgcm experiments. We will attempt to clarify the caption further to alleviate this potential source of confusion.

*"- I don't understand the rationale behind taking a vertical integral to produce the injection rates (L. 179). I can see that for the GOM the freshwater seems to be advected very quickly into the*

*sub-surface as no anomalies seem to be visible at 30m but anomalies are visible at 100m... yet without any clear justification salinity in the top ~100m should be shown and used,"*

The rationale behind taking the vertical integral was to ensure that the salinity anomaly throughout the entirety of the water column was captured no matter the resulting vertical distribution (which, as the reviewer comments on, varies depending upon the injection location). Using only the top ~100m as suggested would be explicitly excluding some components of the salinity anomaly without benefit vs. the approach we've taken which incorporates all the salinity anomaly in the water column and then weights this value on a cell-by-cell basis by the (lateral) integral of the field.

We note that the reviewer, in commenting that '*no anomalies seem to be visible at 30m*' highlights that we need to add comments to some of our figures that the scale is clipped (e.g. Figure 1, where -0.25->0 shares the same colour as values >0).

"*- L157: it states that the simulations are run for 22 to 24 years, but on figure 1 the maximum years shown are 20."*

The longer simulations are those presented in the supplement where OBS configurations were used for FEN and GSL. The text will be updated to focus only on those simulations discussed in the main-text.

"*- It is stated that COSMOS exhibits centennial-scale climate variability under the boundary conditions chosen (L. 198-199). Isn't that a problem for your study? Will the internal variability of the model affect the response to the meltwater injection?"*

The centennial scale climate variations are O(500 years) in scale. The interval chosen reflected a multi-centennial warm interval (i.e. analogous to interstadial conditions) to mitigate the impacts of the internal climate variability on the goals of the investigation. As for this behaviour being problematic here: this configuration, and the internal variability exhibited, reflects our current understanding of the climate system during this time interval. In particular the study focuses on the role of glacial runoff (i.e. meltwater) during this interval. Using a model which exhibits a stable climate during an interval which is well understood to be bi-stable would not reflect reality and thus be of limited utility in this regard.

"*-It would be nice to show the main surface currents in the NA in the control state, maybe on top on of the subplot of figure 3. That would help explain the simulated salinity anomalies."*

As with the MITgcm figures discussed previously, the addition of vectors overtop a spatial field would reduce figure clarity significantly. However, given the resolution of COSMOS is much lower than that of the MITgcm configuration used we will investigate this and update the figures accordingly if figure clarity is not significantly impacted.

"*L. 323 and 325: It is stated that 1dSv hosing results in greater reduction than MAK or FEN, but I think that the 1dSv are actually quite close to the MAK and FEN results. Maybe you simply wanted to say 2dSv?"*

The reviewer missed that the text specifically references 'greater peak reduction' and 'maximum GRIP 2m temperature deviation'. With respect to AMOC, as shown in Fig. 4, the lowest AMOC value for 1dSv of Hosing is ~6.5Sv, while for MAK-R and FEN-R the lowest AMOC values are ~8Sv and

~7Sv respectively. While the values are comparable, the statement(s) in the manuscript are correct. However, we will clarify the language in  these particular sections to address this potential source of reader confusion.

*"L. 328: Why compare after 10 years?"*

This interval was chosen as, generally speaking, this is when the changes were most interpretable. Once the AMOC has transitioned to a stadial state the 2m temperature anomaly variations reflect the climate transition to the stadial state and not the initial transient impact of the localised freshwater injection. We will add a note to this effect in the revisions.

*"L. 331 and elsewhere: It might be good to make a clearer distinction between the different locations of meltwater input and the time of interest. After 50 years, the MAK and FEN results seem similar to the broad NA hosing. However, the GSL and GOM are quite different until at yr 80-100.*

*Does this however mean that if one does not mind an uncertainty of 100 years, then the hosing is fine?"*

The statement from the manuscript is "Thus, the deficiencies introduced by Hosing invalidate its use in comparison to regional injection.". While the results for some fields are similar for the various outlets after some time (injection location depending), this work demonstrates that there is no reason to utilise hosing outside synthetic model benchmark comparisons and in doing so the role of glacial runoff/freshwater-forcing is overemphasised in the climate system, even after 100 years. The argument of 'wait 100 years before making observations' also ignores the role of hysteresis and tipping points in the climate system, where the rate of change or the magnitude of impacts are important (e.g., for the simulation of Dansgaard-Oeschger events). These discussion points will be included in the manuscript to address potentially similar queries.

*"5. The impact on the AMOC of the different hosing locations is probably dependent on the location of deep-water formation in the North Atlantic. As such, the locations of deep-water formation in the COSMOS inter-stadial state should be shown. In addition, the strength of the sub-tropical and sub-polar gyres could impact the advection of salinity anomalies the locations of deep-water formation. The caveat associated with the 2 points mentioned above should be discussed."*

A brief discussion of this, as well as a supplemental figure showing the deep-water formation regions during the interstadial state, will be added to the manuscript.

*"L 429: this is a surprising result, that should be explained. Please also refer to a figure. The more effective salinity crossing is also not obvious from the figures."*

We will review Section 3.1 in light of the changes already suggested by the reviewers and add additional discussion as required. The 'salinity crossing' was difficult to represent  during the investigation, however we will try to add an updated figure to the supplement demonstrating this feature.

*"Throughout: Replace "eddy-parametrizing" with "coarse-resolution" to avoid confusion"*

We will not implement this change as doing so would have the net effect of reducing clarity and precision. 'Coarse-resolution' is a subjective description, which varies significantly depending upon

the spatio-temporal scales a given researcher works within. While 'eddy-paramerterizing' (i.e., not eddy-permitting or eddying) is a definition tied to the scale of a physical value (i.e., the Rossby radius).

*"Throughout: Please avoid starting sentences with "As well, ""*

We will review the manuscript and reduce the usage of "As well" where possible.

*"Abstract: Please define "AMOC""*

Will implement

*"L. 39-40: Climate modelling studies should be cited here"*

As the reviewer does not specify specific studies they wish to be included, and as one of the studies cited incorporates modelling elements as well as observational datasets, we will not add additional citations without utility. However, we will modify the following sentence to read:

"A key uncertainty in the understanding of the effect of freshwater injection into the ocean arises from the fact that most models used to examine the impacts are of insufficient resolution to resolve the eddy-scale features that affect transport and mixing (as detailed below).

*"L. 53: References are needed"*

We will cite, at least,

Dokken, T. M., K. H. Nisancioglu, C. Li, D. S. Battisti, and C. Kissel (2013), Dansgaard-Oeschger cycles: Interactions between ocean and sea ice intrinsic to the Nordic seas, Paleoceanography, 28, 491–502, doi:10.1002/palo.20042.

*"L. 56-57: Remove "for a recent temperature reconstruction""*

We will not implement this change. Removing the requested text would imply the Kindler et. al., 2014 was a primary citation for the GRIP record, and citing instead a primary reference for the GRIP ice core record would be of limited utility as the subsequent interpretation thereof is of more utility to the expected audience of this manuscript. However, we will rephrase to incorporate a relevant reference to the original GRIP dataset and clarify the distinction between it and Kindler et. al.

*"L. 105: "direct hosing" needs to be replaced by something more appropriate and precise."*

Since we have already defined hosing on the previous page we will remove the superfluous word 'direct'.

*"154: Please provide the exact coordinates of the meltwater input locations"*

Will implement

*"216: ppmv"*

We will update the manuscript to switch PPMV and PPBV to ppmv and ppbv as per the CP style guide.

---

## Author Comment (AC2)

**Response to Reviewer (RC2):**

Firstly we would like to thank the anonymous reviewer for the time they have taken to read through the manuscript and make their suggestions. However we do have to refute several of their conclusions, specifically about the applicability of the models, details below. An itemised set of responses to their comments is below, beginning with the more substantial points and technical points following.

With respect to the technical issues the reviewer highlights on the numerical models used:

*"OGCM at 1/6° of resolution, which was already run more than 20 years ago, e.g. New et al. 2021 and the new standard for this kind of ocean-only model is rather around 1/20 to correctly solve the eddies, e.g. Hirschi et al. 2020 among many others)."*

While indeed OGCMs have been run at comparable resolutions in the past there are several flaws with the specific comparison the reviewer makes to both New et., al., and Hirschi et.al.

New et. al., 2001 used $\frac{1}{3}$ of a degree vs. our $\frac{1}{6}$ of a degree, and their setup used fewer vertical layers (20-30 configuration depending vs. 50 as used here). The New et. al., investigation also conducted regional simulations ( 19°S to 70°N) vs. our global scale simulations. Assuming New et. al.'s zonal range (which is unspecified) is 260E to 360E we find they have a total of 1,602,000 grid points in their 267x300x20 configuration (assuming 20 layers and a uniform $\frac{1}{3}$ degree lat-lon grid) vs. the 78,030,000 of our 510x510x6x50 (the 510x510 cubed-sphere configuration). Not only does this represent a significant difference in run-time memory, output data storage, and process interconnect (i.e. message-passing-interface, MPI) traffic requirements from a purely technical perspective, we also have to use a smaller timestep to satisfy Courant–Friedrichs–Lewy (CFL) stability. In summary, this comparison is not accurate nor constructive.

Both New et. al., 2001 and Hirschi et. al., 2020 (the reviewer having specified only those two references we restrict our comparisons to these same references) are contemporary background climate modelling investigations. At present, to the best of these authors' knowledge, the manuscript presented uses one of the highest resolution glacial interval OGCM modelling simulations to-date. This configuration is surpassed only by the AWI-ESM model, as in Shi et. al., 2023, whose ocean model uses a finite-element approach and thus has some regions of higher resolution (depending upon the configuration used). There are significant difficulties in applying GCMs, let alone coupled Earth systems models, configured for present day or preindustrial climates to glacial time intervals as well as different scopes of work (chiefly, the longer millennial scale integration times required for paleoclimate investigations vs. that of centennial scale CMIP investigations) that the reviewer highlights, but then ignores within the context of the current manuscript.

*" Thus, we cannot really say that the tools used are really state of the art."*

With respect to this statement in the context of the MITgcm, we argue that this has already been recently highlighted (Love et. al., 2021) and refuted in past work using this model configuration in https://doi.org/10.5194/cp-2021-15-AC1 as part of the discussion of Love et. al., 2021. The specific text is reproduced here for clarity and posterity.

"We point out the main focus of this work, the representation of surface transports and features generally regarded as subgrid scale, would not benefit from existing updates to the model as the features of interest are already adequately represented in the version we use. Updates to the MITgcm model appear to largely center around bug-fixes and documentation updates

(https://github.com/MITgcm/MITgcm/releases) without substantial effect on the representation of surface transports and eddies. As well, with regards to increasing the resolution of ocean-only simulations, we do note there are some entries in Hirschi et al. (2020) (which for the benefit of those unfamiliar with the work, is a review paper examining the representation of AMOC under present-day conditions from multiple sub 1 degree resolution model simulations extracted from 23 different publications) which are higher resolution. However, only one is a global ocean-only simulation which is above our grid resolution (Moat et al. (2016) which used 1/12 degree). Thus, we contend that the model configuration used in this study is of comparable complexity and resolution to the multi-model ensemble of simulations presented in Hirschi et al (2020). We make this point in the revised submission."

With respect to COSMOS, the authors evaluated the most recent versions of the component models (tagged as MPIESM-P in PMIP4/CMIP6) as part of the preliminary work for this and other parallel investigations. The primary difference between these configurations, barring resolution which was a choice informed by multi-millennial scale simulation requirements and compute cost, is the updated version of the atmosphere component from ECHAM5 to ECHAM6. MPIOM was updated as well, however these updates were primarily 'quality of life' updates addressing I/O difficulties with the model. Having evaluated the additions (and associated computational expense equalling an increase of roughly a factor of 2) to ECHAM6 we found the trade-off in model expense to not be worthwhile for the goals of our investigations. We also point out that the reviewer is incorrect in classifying COSMOS as an AOGCM, as that neglects the land surface model JSBACH. While output of this model component was not included in this investigation (as it will be the subject of future work) we argue that describing COSMOS as an Earth Systems Model is more accurate.

*"Models with such coarse resolution are now still used since they can allow to perform long transient runs (multi-millennia) to analyse intrinsic AMOC variability during glacial time for instance, but here only simulations of 100 years or less are performed."*

What the reviewer is unaware of, as it was not communicated in the paper since these are technical details (and in the authors' view outside the scope of the manuscript), is that the COSMOS simulations presented here form only a small subset of sensitivity experiments comprising >10,000 simulation years investigating the roles of glacial runoff and pCO2 in centennial to millennial scale climate variability. Each of the simulations presented have been integrated for several thousand years beyond what is analysed in the manuscript. However, the freshwater forcing fluxes used, which while comparable to the which we can infer from the proxy record and why they were used in Love et., al., 2021, are strong enough to suppress overturning in COSMOS and results in a stadial state. As such, and as discussed in the response to the other reviewer, the resulting climate changes from the stadial state prevent any useful comparison beyond the window presented in the manuscript.

*"Since this study aims at evaluating climate dynamics during paleo-periods, we can wonder if such short-time simulations are really relevant to improve our understanding of climate dynamics as compared to existing records."*

We note that in-development work within our research group has found that the triggering of large scale (with respect to both spatial and magnitude) centennial to millennial scale climate variability (i.e., Dansgaard-Oeschger-like events) is sensitive to climate perturbations and internal variability on sub-annual timescales. As such, the scales presented here are of at least a temporal scale to be relevant for such investigations.

*Abstract: It is quite long and not very clear. I agree that the experimental design used is quite complex and it took me time to understand it, but please try synthesize better what you've done and why (and leave details out of the abstract to focus on the main results).*

We will attempt to clarify the abstract.

*• Line 68: you can consider to cite Swingedouw et al. (2022) to substantiate this claim (although this was using an eddy rich model).*

Assuming the statement in question is "This justification for hosing is … important to the transport and mixing of freshwater."

We will cite at least the requested manuscript.

*• Line 75: the link with climate change is not obvious given that the focus here is on glacial time and paleostudies. Please state this more clearly.*

We will add additional, specific, examples of the link between freshwater forcing methodologies (specifically studies reliant on Hosing) and climate change.

*• Line 88: unclear sentence? What was missing? Please be more explicit.*

The statement commented on is "Lohmann et al. (2020) provides a mix of eddy-parametrizing and eddy-permitting conditions through their use of an unstructured mesh approach with the FESOM model."
We are unsure what action the reviewer is requesting here, as subsequent sentences describe and compare the work of Lohmann et. al., 2020 within the context of other work discussed within the introduction. However, we will try to rephrase to increase clarity.

*• Line 108: "cannot" is too strong. The parametrizaiton aims at reproducing that. What you mean here is that they are not doing a good job at this, which I agree, but maybe with better parametrizaiton this can be solve*d.

We will modify the sentence to
"Given that eddy parametrizing models do not accurately simulate the pathway of coastally released meltwater, …"

*• Line 121-132: As far as I understand the use of MITgcm is only for finding the fingerprints that are then used in the coarse resolution OAGCM. This is not stated very clearly here, while the title of 2.1 states implies it but not that clearly. Please rephrase a bit to clarify this.*

We will rephrase the first paragraph of Section 2 while avoiding repeating the first sentence of Section 2.1 which reads "All of the fingerprint generation simulations are performed using the MITgcm …" as reviewer #1 has requested that we remove repetition.

*• Line 134-150: not much is said about the mean state of this MITgcm run. What is the strength of the AMOC, where are located the convection sites. Is the circulation realistic for glacial time?*

The MITgcm run(s) are described in Love et. al., 2021 and this is stated in the manuscript. Given we utilise the MITgcm over a short interval and extract only the salinity anomaly any discussion of the MITgcm's AMOC is of no utility to the reader. As for whether the circulation is realistic, the authors are unaware of any proxy data which can be used in a robust model-data comparison to evaluate the realism of surface ocean circulation during the last deglacial interval. Will will be providing plots showing surface circulation as requested elsewhere. We can at least state that the resulting distribution of glacial runoff for the Mackenzie River outlet compares favourably with previous results discussed in Condron and Windsor, 2012 within the limitations of comparing different experimental setups with the same model.

• *Line 165: replace "is" by "might be", since the use of only an eddy-permitting model is really questioning the realism of the results.*
We disagree and will not implement this change given the discussion above regarding the relative resolution of the MITgcm configuration presented here vs. the state-of-the-art for glacial ocean simulations as well as the 'proof-of-concept' nature of the fingerprint component of the investigation.

• *Line 174: when are those 5 years selected (what time of the simulation).*
The line will be modified to read

"… averaged over the last 5 years of each model simulation."

• *Line 181: this claim is not supported by any figures I think, so a "not shown" is necessary here.*
Will Implement

• *Line 198: if there are millennial intrinsic oscillation in COSMOS, how did you change the period of the control simulation? Is the AMOC changing on the long term?*

With respect to 'how did you change the period of the control simulation', we are unsure exactly what the reviewer is requesting here. When executing the model we simply specify the year from which the model components are restarted and integrate the model forward. With respect to 'Is the AMOC changing on the long term', there is no long-term secular drift in the AMOC present on multi-millennial scales in the control simulation. If the reviewer is inquiring as to how we selected the control interval we did: we chose the interval with the longest interstadial duration that was at least 1000 years past the change to the background conditions in the model setup.

• *Line 210: since the focus here is on the freshwater release, the way this freshwater is accounted for in the model should be depicted. Is the model rigid lid or free surface? Which kind of parametrization? Is salt conserved with this parametrization?*

The freshwater is injected into the ocean model via the same field by which liquid precipitation is passed from the atmospheric model via the coupler. The model uses a free-surface formulation. This information will be added to the manuscript.

• *Line 213: still this question about intrinsic variability during glacial time?*

We are unsure exactly what the reviewer is requesting here. The sentence in question is 'The control simulation used here has been integrated forward for roughly 1000 years under the current climate con-
ditions prior to initializing the injection simulations.'. This scale of model simulation time is typical in paleoclimate modelling investigations to allow for any adjustment in the model mean-state to the modified boundary conditions.

• *Line 249: within those questions, there is an implicit assumption that the response of model to freshwater only depends on their pathway, which is totally false, as shown in Stouffer et al. (2006) for the same design of freshwater release, the model responses can vary by an order of magnitude in terms of AMOC!*

The reviewer seems to be implying that in the stating of our research questions we are neglecting that the investigation is a modelling study and presenting results for those models which are used (and introduced in the preceding section).This is a critique that could be levelled against most climate model based studies to date. We will add an explicit caveat on model dependence to freshwater forcing especially with respect to climate response.

• *Fig. 5: I do not find this analysis very enlightening: the curves are very messy and not much is said about the differences. Are they due to AMOC response? A correlation of the climate response this AMOC response (with dfferent lags) might be the least to be done. The residual might represent the effect of other processes than change in meridional heat transport due to the AMOC, including atmospheric noise, change in stratification, gye transport, etc.*

We will attempt to incorporate some additional statistical analysis regarding the correlation and lag/lead of the presented climate metrics vs. AMOC. If the additional analysis is informative and provides useful results then they will be incorporated into the manuscript, if they are not then this finding will also be summarised. With regards to the lines being messy, the data already has a 10-year running mean applied to it as documented in the manuscript, and the level of variability presented is not atypical of other published modelling investigations using coupled climate models e.g. Klockmann et. al., 2018, Klockmann et. al., 2020, and Zhang et. al., 2021.

• *Line 389: unclear sentence.*

The sentence in question is "In spite of this result, the use of fingerprints results in a freshwater distribution more dis-similar to the eddy-permitting simulations than the regional injections for all injection sites considered."

This sentence summarises a small set of comparisons presented in Section 6 of the supplemental materials. We will rephrase.

• *Line 404: this is "not shown", is it?*

The site of the GIN seas deep convection is indeed not shown in any of the Figures at present but is mentioned in the model description on lines 207-208. Such figures have been requested elsewhere and so we will cross reference the appropriate figure here.

• *Line 411; any correlation to support this claim?*

Contemporary investigations such as Cosimo, 2002 and Shu et. la., 2012, have found similar correlations between surface air temperature and sea ice concentration at both poles. We will add additional citations and some physical reasoning to provide additional context to the highlighted statement.

• *Line 417-419: unclear sentence.*

The sentence in question reads "Comparing regional injections to their respective fingerprint counterparts, injecting freshwater at the outlets results in earlier, but not faster, SIA/SST/SSS changes in the GIN Seas region for the MAK and FEN regions relative to the fingerprint methods." We will modify to read

"Comparing regional injections to their respective fingerprint counterparts (i.e. MAK-R to MAK-FP), ..."

• *Line 435: replace "is" by "might be" since this is a hypothesis at this stage.*

We will modify the text to

"This is likely due to the stronger..."

• *Line 440: "first principles". What do you mean? Please be more specific here.*

First principles here being: If you inject large quantities of freshwater directly over regions of deep water formation, or directly adjacent to such regions, you will of course reduce the strength of density driven convection. We will rephrase to "buoyancy considerations" for clarity.

• *Line 445-447: please rephrase, I do not get the reasoning here.*

We will rephrase to "This ordering is reversed (i.e., GSL sourced freshwater results in greater impacts than GOM) and results align ..."

• *Line 450-451: Fig. S7 should show significant results only! (using a t-test of the difference for instance).*

We will provide plots of the standard deviation of the field in the control simulation over the averaging duration to provide a comparison of the anomalies relative to the variability in the field.

• *Line 465-475: since the site of convection are not shown (among many other things, e.g. barotropic streamfunction, etc.), I think that the analysis is in the end quite poor and not going much into depth of real understanding of the differences in response.*

As discussed elsewhere, we will add plots showing the sites of deep water convection and expand upon this subject in the manuscript.

• *Line 481-483: where is it shown?*

We assume the reviewer is referencing the following on lines 480-481 (as otherwise the reviewer's comment does not make sense). "The differences are largest at the location of the source of glacial runoff"

We will add references to figures 1 & 2.

**References**

Comiso, J. C. : Correlation and trend studies of the sea-ice cover and surface temperatures in the Arctic, Annals of Glaciology. Cambridge University Press, 34, pp. 420–428. doi: 10.3189/172756402781818067, 2002

Condron, A. and Winsor, P.: Meltwater routing and the Younger Dryas, Proceedings of the National Academy of Sciences, 109, 19 928–19 933, https://doi.org/10.1073/pnas.1207381109, 2012.

Klockmann, M., Mikolajewicz, U., and Marotzke, J.: Two AMOC States in Response to Decreasing Greenhouse Gas Concentrations in the Coupled Climate Model MPI-ESM. J. Climate, **31**, 7969–7984, https://doi.org/10.1175/JCLI-D-17-0859.1, 2018

Klockmann, M., Mikolajewicz, U., Kleppin, H., & Marotzke, J.: Coupling of the subpolar gyre and the overturning circulation during abrupt glacial climate transitions. Geophysical Research Letters, 47, e2020GL090361. https://doi.org/10.1029/2020GL090361, 2020

Love, R., Andres, H. J., Condron, A., and Tarasov, L.: Freshwater routing in eddy-permitting simulations of the last deglacial: the impact of realistic freshwater discharge, Clim. Past, 17, 2327–2341, https://doi.org/10.5194/cp-17-2327-2021, 2021.

Shi, X., Cauquoin, A., Lohmann, G., Jonkers, L., Wang, Q., Yang, H., Sun, Y., and Werner, M.: Simulated stable water isotopes during the mid-Holocene and pre-industrial periods using AWI-ESM-2.1-wiso, Geosci. Model Dev., 16, 5153–5178, https://doi.org/10.5194/gmd-16-5153-2023, 2023.

Shu, Q., Qiao, F., Song, Z. et al. Sea ice trends in the Antarctic and their relationship to surface air temperature during 1979–2009. Clim Dyn **38**, 2355–2363, https://doi.org/10.1007/s00382-011-1143-9, 2012.

Zhang, X., Barker, S., Knorr, G., Lohmann, G., Drysdale, R., Sun, Y., Hodell, D., and Chen, F.: Direct astronomical influence on abrupt climate variability, Nature Geoscience, 14, 819–826, https://doi.org/10.1038/s41561-021-00846-6, 2021.